# Mechanical Behavior of Calcium Sulphate Modified with Citric Acid and with Added Carbon Fibers

**DOI:** 10.3390/polym14081522

**Published:** 2022-04-08

**Authors:** José Antonio Flores Yepes, Luis Miguel Serna Jara, Juan Manuel Berná Serna, Antonio Martínez Gabarrón, Ana Maria Codes Alcaraz

**Affiliations:** 1Engineering Department, Miguel Hernandez University (U.M.H.), 03312 Alicante, Spain; ja.flores@umh.es (J.A.F.Y.); juan.bernas@umh.es (J.M.B.S.); antonio.martinez@umh.es (A.M.G.); ana.codes@goumh.umh.es (A.M.C.A.); 2Faculty of Science and Technology, Universidad Isabel I (UI1), Fernán González Street 76, 09003 Burgos, Spain

**Keywords:** gypsum, additive, bending, compression, carbon fibers, citric acid

## Abstract

The study and subsequent analysis of the interaction of calcium sulfate with added citric acid and with two additional proportions of carbon fibers of different lengths has been based on the IMR and D Method for its realization. The purpose of this work is the study of the physical and mechanical behavior of the resulting material between the intimate mixture of calcium sulfate with additives and carbon fibers, justifying said work with a link to the Sustainable Development Goals (SDG) regarding the benefits that the Calcium sulfate has contributed to civil society since times dating back to ancient Egypt. We find ourselves with a material of which the energy used in its manufacture is far from that required by steel or cement, and construction with this new compound is in a much higher stage than construction with adobe. Therefore, this is a compound that can be developed for a wide variety of applications. The novelty of this study is the inclusion of polymeric fibers in a material used over the centuries to improve its mechanical properties. With these improvements we will be able to reduce thicknesses in manufacturing, which implies a reduction in manufacturing energy and weight structures in buildings, which should be studied and analyzed in the future. The kneading of calcium sulfate with long fibers at high percentages complicates not only the results, but also the manufacturing process. As representative results of the study, we can indicate that a composite material with high mechanical capacity has been achieved, with maximum values of flexural strength of 8.12 N/mm^2^ and compression strength of 17.58 N/mm^2^.

## 1. Introduction

Man is consuming perishable resources that have reached a point of no return since the system is not sustainable. This includes energy resources (oil, gas, electricity production, etc.), which in turn produce pollution while being used or manufactured. This has consequences for the environment and climate change.

One of the objectives has been (and continues to be) recovering the use of calcium sulfate, as it is a material that has widely used in construction since time immemorial. This material allows us infinite possibilities and constitutes an intermediate state between the economy of means (manufacturing with stone or adobe and modern construction (with cement/concrete and steel)).

Gypsum, as a material used in construction, has been analyzing the setting mechanism for some time [1,2,3,4], as well as the workability and physical behavior of the plaster mortar when it is added.

There are also numerous works using additives that improve or modify some of its properties; for example, the use of glucose, citric acid, and sodium bicarbonate to retard hydration [5]; or the use of melanin formaldehyde in the manufacture of plasterboard [6]; to meet the minimum resistance requirements required by the standard [7]; or the use of graphene powder as an additive to improve its mechanical properties [8].

The additives used with the plaster are used to improve outdoor durability. The authors of [9] added hydrated lime and a very fine aggregate of crusher dust to increase said resistance; Feng-qing [10] used granulated blast-furnace slag, ashes and cementitious additives to obtain water-resistant plasters; and Li [11] used a complex water-repellent gypsum additive intended to change the water resistance based on its microstructural properties.

Schneider’s study [12] on the effect of retarders focuses on the configuration process in gypsum hemihydrate, using the additives citric acid and tartaric acid. In this case, the results of mechanical resistance (the object of this work) are not reflected in this study compared with other additives.

There are published studies [13] on the effect of additives on the mechanical properties of plasters produced by combustion gases compared to natural plasters. In this case, in addition to the microscopy, only the flexural strength is analyzed, using sulfuric acid, potassium sulfate, acetic acid, citric acid and methylcellulose, achieving flexural strength of around 6 N/mm^2^ with 5–8 min of setting time.

In another study, Merino [14] also studied the effect of adding glass fibers to the gypsum matrix using a superplasticizer in its base, indicating that the use of these additives improves and favors the inclusion of addictions by reducing the water in the setting matrix and increasing its time. Marcos Lanzón [15] studied different concentrations of citric acid and evaluated its mechanical behavior, as well as its microstructure and setting time.

Other works dealing with the effects of the use of additives systematically focus on the setting structure in crystal formation, as well as the influence of some additive bases on the general behavior of the microstructure [5,13,14,15,16,17]. Other researchers, such as Wu [18], studied the effect of the addition of polyacrylic ester emulsion on the mechanical properties of desulfurization gypsum. The result shows that 1% by weight of the addition leads to a compressive strength of 46.9 N/mm^2^ in the plaster; for its part, the author of [19] used self-leveling mortars based on phosphoric gypsum (alpha-Hemihydrate gypsum, α-HH), analyzing the effect on the setting time of three retarders: Protein Salt, Citric Acid and Sodium Tripolyphosphate. The flowability was individually investigated using three different types of superplasticizer, namely naphthalene, polycarboxylate and melamine, obtaining delays of 80 min and corresponding compressive strengths of 54.25 N/mm^2^, 53 N/mm^2^ and 52.25 N/mm^2^.

The developments in recent years have allowed us to obtain interesting results, as well as the availability of its use in many applications. In this line, some of its properties have been improved, including mechanics, manageability, water content, hardness, as well as elasticity and fire behavior, in addition to its intrinsic properties of being fireproof, thermal, acoustic and thermo-hygrometric [20].

Gypsum is mainly made up of calcium sulfate with two water molecules, which is called calcium sulfate dihydrate or dihydrate. One of the most widespread uses of plaster is in construction, due to the aforementioned properties [21]. It is a material with extensive solvency and easy handling.

The use of composite materials is growing rapidly, establishing itself in a wide variety of industrial sectors. This is due to its excellent mechanical properties, obtaining light, ductile materials, mechanical strength and resistance to high temperatures [22]. Carbon-fiber composite materials in a polymeric matrix currently have a wide field of applications in the aeronautical, automotive and medical industries [23]. In recent years, carbon fibers have been used in different applications, where their mechanical properties and lightness are highly taken into account. Likewise, polymeric matrix composite materials are increasingly used since they have excellent mechanical properties with respect to a high strength-to-weight ratio [24].

Thus, the main objective of this work is to study the use of carbon fiber as an additive to calcium sulfate mortar, analyzing its mechanical behavior in different lengths and proportions, with the following properties: shore surface hardness, density, flexural strength, compression, elasticity and microscopic matrix. As Ramos, J. [25] comments, carbon fibers as a reinforcing material reflect greater effectiveness in deflecting internal fractures in composite materials, ensuring that they do not follow a defined path, and, in turn, that they achieve excellent mechanical properties by having the ability to absorb energy.

## 2. Materials and Methods

### 2.1. Materials

The materials used to carry out the test, as well as the equipment and instruments used, are listed in this section.

#### 2.1.1. Water

Water is one of the basic components required for the formation of the paste. The water used in our tests was obtained from the supply network, without any special treatment. The temperature at which it is added must be taken into account, since this affects the amount of calcium sulfate that dissolves. The laboratory temperature here was 25 °C. Table 1 shows the analytical parameters of the water used in the laboratory tests for the different batches.

#### 2.1.2. Plaster

Brown plaster (construction plaster B1) and water obtained directly from the drinking water network were used, to which an organic-type additive was added. It had a fluidizing and retarding effect on the setting process. Added carbon fibers varied both the percentage and length.

The calcium sulfate used for the development of the tests, whose nominal characteristics are defined according to the EN UNE 13279-1 [7] standard (Table 2), is called type B1 Plaster for construction. It is thick plaster consisting of hemihydrate (SO_4_Ca.1/2H_2_O). It is more than 50% hemihydrate, with a minimum mechanical resistance to Flexion of 1 N/mm^2^ and 2 N/mm^2^ to Compression (Table 1). This type of plaster is used for bonding paste in the execution of partition walls, in interior linings, as an auxiliary conglomerate for construction, and is also used for prefabricated elements.

#### 2.1.3. Additives

The organic acid, which retards setting and improves workability, is a 25% solution of citric acid. As has been indicated, it has a retarding effect on the setting process, being able to handle the mixture long enough for the manufacture of test pieces and subsequent analyses. A similar additive can be used in solid or liquid form after dissolution in water and can also be mixed with other organic acids such as acetic or tartaric. The properties of this additive are shown in the following Table 3.

The other additive used in laboratory tests, in combination with citric acid, is carbon fiber. This material is supplied in coils of wire; it is fundamentally used for reinforcement in high-performance composite materials. Subsequently, for the preparation of samples to be tested in the laboratory, it will be cut according to the consigned test sizes. The general properties of this material according to its technical specifications are shown in Table 4.

The sizes used (lengths), as well as the percentages by weight, are established in Table 5. This is referenced for the mixture dosage, which is 2500 g of plaster per batch. This is necessary to formalize the 9 test tubes (160 mm × 160 mm × 40 mm) that refer to each one of the batches (3 minimum for each sample); the fibers were randomly oriented.

### 2.2. Methods

We had two different test groups because as the length of the carbon fiber increases, the kneading requires more water; therefore, we needed two references to compare results.

The first sample of calcium sulfate kneading without carbon fiber maintained a water–gypsum ratio (W/P), in reference to weight, in the proportion of 0.5 (50 g of water per 100 g of calcium sulfate). This was recorded as Y1. The second sample had a water–gypsum ratio (W/P) of 0.6 (60 g of water per 100 g of calcium sulfate). This was recorded as Y2. For the first reference test (Y1), it was added with 0.02% on the basis of calcium sulfate. That is, 0.2 g per 1000 g of calcium sulfate. The second test (Y2) increased the amount of additive to achieve a little more fluidity and handling time (necessary due to the high pastiness of the mixture that makes handling difficult), using 0.06% on calcium sulfate; 0.6 g per 1000 g of calcium sulfate.

In relation to the number of batches and test tubes required, the criterion of the UNE-EN 13279-2 Standard [26] was maintained. Three batches were made per test, and each of them contained 8 samples of 40 mm × 40 mm × 160 mm, according to the length and percentage of fiber. Thus, it was calculated that the necessary amount of plaster to make a batch consisting of 8 specimens was 2500 g. For tests C1 to C8, whose fiber length was shorter, a dose of 50% of the amount of gypsum was used; that is, 1250 g of water. For the following tests, C9 to C16, (with higher percentage and fiber length), the water was increased to 60%; that is, 1500 g of water per 2500 g of gypsum.

The following Table 6 indicates the percentage of additive used, as well as the dosage used for each batch of test tubes.

The mixtures of plaster, additive and carbon fibers were mixed using a Lödigue brand mixer, obtaining a homogeneous mixture that depends mainly on the length of the fiber.

Once the mixtures were obtained, they were kneaded manually (water temperature 23 ± 2 °C) and they were poured onto the standardized molds. The standardized specimens were obtained to later introduce them in a drying oven. Once the mixtures were obtained, they were kneaded manually (water temperature 23 ± 2 °C) and they were poured onto the standardized molds, and the standardized test tubes were obtained to subsequently introduce them in a drying oven. After seven days (maintaining the drying conditions with RH 50 ± 5%), they were tested in a universal testing machine to obtain the resistance and modulus of elasticity and density of the specimens by flexotraction, automatically. Said test was carried out according to the test protocol established in the UNE-EN 13279-2 Standard [8].

To know the values of resistance to surface penetration or hardness, the Shore method was used, with a C scale device: “Baxlo brand”. A portable hardness tester was used that has a sensitivity of 5 shore units, with a capacity of 0 to 100 shore units. The compression tests were carried out with a testing machine for this purpose, adapted for 40 mm × 40 mm × 40 mm specimens. “Brand: Controls”.

After testing the specimens from the different batches made, Microscopies were carried out using a Scanning Electron Microscope model “SIGMA 300 VP from ZEISS” with Schottky hot cathode field emission. The equipment has secondary electron (SE) and backscattered electron (BSE) and X-ray (EDX) detectors, as well as an inLens detector for secondary electrons and one for transmitted electrons (STEM).

Next, in Figure 1, the flowchart of the methodology carried out in the experimentation is shown.

## 3. Results

The results obtained can be seen in Figure 2. We observe the results of the breakage of the test specimens, including how visible cracks appear on the surface, which increases with the fiber content provided.

The values of the different tests are shown below in Table 7. These are the reference values of the test Y1 and Y2, as indicated for each group of batches identified with the same amount of additive and water. Values with the same letter in the same column are not significantly different, according to Duncan’s multiple range test at the 0.05 level of significance.

## 4. Discussion

Below are the comparative tables in relation to the length of the cut and the percentage used in the different tests. This comparison will allow us to determine what percentage and length of fiber obtains a better result from the mortar, and although it is evident that the most important parameters are the increase in mechanical resistance that implies the extension of the use of calcium sulfate to other applications, the complexity in manufacturing and the influence that the fiber has on its dispersion through the matrix will also be evaluated.

We start by looking at Shore C hardness.

In general, as shown in Figure 3, the surface hardness varies from the first group (C1 to C8) with respect to the second (C9 to C16), due to the influence of the mixing water. As the water increases, the hardness decreases significantly. The test reference Y1 (89.96) and Y2 (86.53) already observed that they do not contain carbon fibers. On the other hand, the deviations in the tests depend on the fact that, when carrying out the measurement, there are certain “holes” in the kneading caused by the fibers.

The mean of the first group, including Y1, is 89.66, while for the second it is 87.18. Even so, high surface hardness of the matrix seemed to imply a closed capillary network. Regarding the tests, the highest hardness was obtained in test C4, with a value of 92.9. This result may be due to the measurement pointer touching the carbon fiber part. Let us remember that this test corresponds to 1.5% of added fiber (37.5 g) and 12 mm in length. The horizontal reference of the additive-free gypsum remains at an average value (in both groups).

### 4.1. Analysis of Resistance to Bending and Compression

As shown in Figure 4, one of the initial assessments is that the carbon fiber, due to its orientation [9], turns the mixture into a composite material; in this way, the fibers act as a reinforcing material, helping to distribute the stresses at 45° in addition to the corresponding traction.

In all trials and in both groups, the same pattern of behavior was observed. The resistance increased with the percentage of added fiber until falling in the last part; that is, when we reached the 1.5% contribution. This is due to the fact that the dispersion of that amount of fibers in the plaster matrix is not very effective. In the second group, in which we worked with more water because a greater length implies less precise dispersion, this did not seem to affect the result; on the contrary, we obtained greater resistance by increasing the length of the fibers. We compensated for the decrease in resistance due to water with the size of the carbon fiber. Note that, for the first group, the incorporation of fiber implies an improvement of 140%; that is, we went from 5.83 N/mm^2^ to 8.17 N/mm^2^. In the second group, from 5.14 N/mm^2^ to 9.38 N/mm^2^, it implies a 182.49% increase. With the exception of test C8, all remained above the reference. This decrease is clearly explained when we observe the microscopy. This is due to the eggs that originate from a lack of uniformity in the mixture because the percentage of 1.5% is excessive.

For the compression tests, the same behavior was observed (Figure 5). Uniform growth as the amount of fiber increased until falling with the proportion of 1.5% and even with that of 0.75% in some tests, reducing the percentage that would have to be provided in the event that the material was requested in compression rather than bending.

In this case, group 2, whose average was 13.35 N/mm^2^, is not at the same level as group 1, which reached a compression of 15.03 N/mm^2^ on average; that is to say that the water here did have a negative influence, despite the fact that in the C15 test, it reached the maximum value of all the tests, reaching 17.58 N/mm^2^. The C8, if it maintains the equivalence with the bending test, stands out as the worst of all, below the reference average. It is shown that group 2, with greater lengths, is not conclusive in the stable behavior of the composite material.

### 4.2. Density and Modulus of Elasticity

There is a clear and evident trend (Figure 6). If carbon fiber is added, the material will weigh less. If a lighter material is sought, it is achieved by increasing the amount of fiber; but, as we have already observed, this is not in line with the mechanical properties, since it makes kneading difficult, and internal voids will be created that reduce its properties. In the kneading, the fibers entangle with each other, creating compact packages (skeins) that are isolated from the calcium sulfate.

In relation to the modulus of elasticity (Figure 7), the greatest deformation is obtained with small values of carbon fiber, with the percentage of 0.25% and 0.5% being the highest value in the C5 test, precisely with 0.25% and fiber length of 25 mm. The pattern of behavior is similar; as the amounts of fibers in % and lengths increase, the results worsen.

This is due to the uniformity of kneading that we achieve when the quantity is small. Trial C8 continues to appear as the worst of all below the reference value. However, the additive-free plaster has above-average behavior, thus obtaining results similar to the work of Nava-Gastelum [27], in which the contribution of carbon fiber in the polymeric matrix helped to reduce the percentage of deformation, preventing the composite material from reaching a complete rupture.

In other comparative studies where the use of carbon fibers in concrete beams was carried out, it was also observed that this provided a better use of the resistant capacity [28], as in the study by Nava-Gastelum [27] on carbon fibers in epoxy resin where an increase in resistance was also seen as a result of this.

### 4.3. Micrograph Analysis

For the analysis of the microscopic structure, the samples were coated with chrome. They were analyzed in the SIGMA 300 VP scanning electron microscope. As can be seen in the following Figure 8a,b and Figure 9a,b, the test references Y1, with a ratio A/Y 0.5; that is, with 50% water, and with an additive proportion of 0.02%, and Y2 with 0.06%.

Regarding the size and formation of crystals, no great quantitative differences were observed when observing the microscopy, although, regarding the size of the grain (of the crystals), these increased with the setting time; see the case of sample Y2, which also maintains less void formation in its structure, in contrast to Y1, whose reaction rate is higher and provides a finer grain.

Let us observe the result in Figure 10a–e, which depicts the carbon matrix at different depths of the field of view.

We can clearly see how the carbon is bonded between the calcium-sulfate matrix. (The samples correspond to the first dosage of 0.02% additive.) The groups of carbon fibers remain visible; these are organized in packets and do not disperse as easily through the calcium-sulfate matrix. In addition, spaces (hollows) appear that originate due to the interference of carbon fibers in the kneading. These gaps (increased by the quantity and length of the fibers), cause a decrease in mechanical properties by increasing both the percentages and the lengths. A priori, it may be thought that the water could be a dispersant for the mixture, improving the compound, but as is observed with the reference values Y1 and Y2, the increase in water acts significantly in the reduction of properties that must be compensated with the carbon fiber.

This is based on the idea that small sizes and percentages give a better distribution of carbon fiber on the calcium-sulfate matrix.

### 4.4. X-ray Chemical Analysis

A general analysis with a scanning microscope, and high vacuum, using the X-ray dispersive energy technique, indicates the following chemical compositions, as reflected in Figure 11.

The X-ray spectrum shown in Figure 12 shows us the following scheme according to the type of transition of the electrons, K being the type of transition from the highest layer to the innermost. Chromium, as already indicated, is part of the coating material of the sample and is not part of the composition.

Now we focus the measurement (microscopy and X-ray analysis) on a specific area, which is recorded with a marked measurement point, for the case of carbon fiber (Figure 13, Figure 14, Figure 15 and Figure 16).

## 5. Conclusions

With these tests, the most relevant conclusions we have reached are as follows.

A composite material with high mechanical capacity was achieved, with maximum values of flexural strength of 8.12 N/mm^2^ and compression strength of 17.58 N/mm^2^. The composite material is more homogeneous and stable with small percentages of fiber, as well as lengths (12–25 mm and 0.25–0.5%), and it is necessary to avoid going outside this range unless some type of fiber dispersant is used. At the time of its manufacture, the kneading of long fibers and high percentages complicates not only the results but also the manufacturing process of the mixture.

In relation to the plaster with additives, the percentage of mechanical resistance to bending is increased by 182.49% (5.14 N/mm^2^ to 9.38 N/mm^2^). Regarding compression, an improvement in resistance of 144.33% (12.18 N/mm^2^ to 17.58 N/mm^2^) is achieved.

It has been shown that the mixture of plaster with carbon fiber in different proportions gives the new material resulting from the mixture greater mechanical properties for use in different uses, such as material for cladding vertical walls and horizontal in construction. It is known to be a material of which the energy required for its manufacture is far from that required by steel or cement, with consequent energy savings and reduction in its carbon footprint. As a result, it is produced in large quantities in Europe, and is widely used in various applications, in industry in general and more specifically in construction, in decorative elements and in medicine, traumatology and dentistry.

Comparing the results of the different additives used in these tests with other innovative additives that can be found today on the market, such as graphene powder, which can be used as an additive in plaster [8], it can be observed that the flexural strength values of the tested additives are higher than those obtained with graphene powder as an additive to plaster, and much higher than those required by European regulations [7], which state that flexural and compressive strengths must be greater than 1 N/mm^2^ and 2 N/mm^2^, respectively, for construction plasters.

It was shown that the mechanical properties are increased by using carbon fiber in the mixture with plaster. When using this mixture in coatings for construction, there will be a reduction in the thickness of the gypsum lining when increasing its properties of resistance to bending, hardness and resistance to compression, thus reducing the quantity of plaster mortar to be used. This generally implies a reduction in manufacturing and energy costs in its production, whose quantification will be carried out in future studies.

As a future recommendation, a new line of research is proposed on the study of the energy quantification of the manufacturing process, the energy savings that are produced, and the reduction of carbon dioxide emissions into the atmosphere.

## Figures and Tables

**Figure 1 polymers-14-01522-f001:**
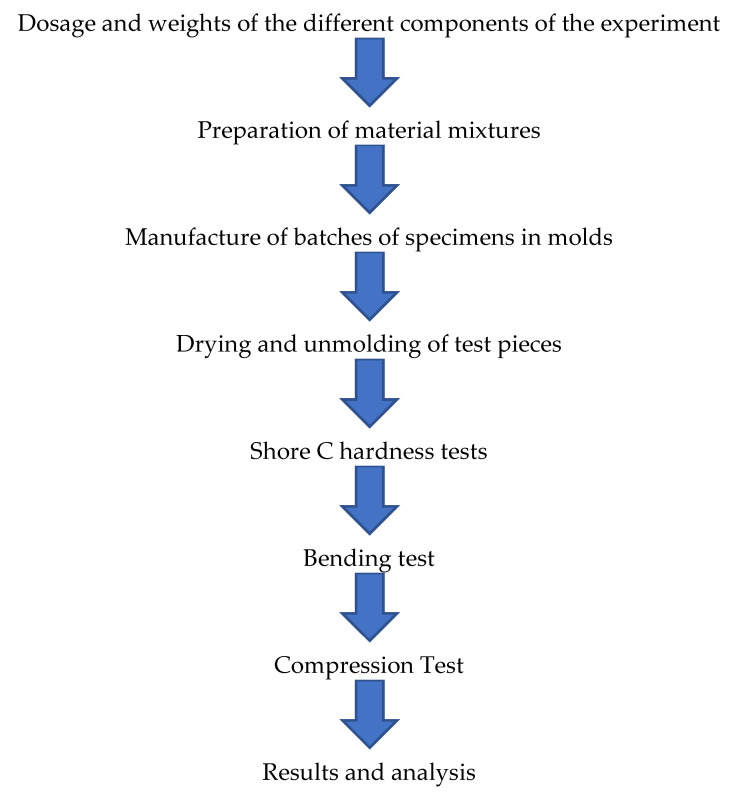
Flowchart of the experimentation methodology. (Source: the author.)

**Figure 2 polymers-14-01522-f002:**
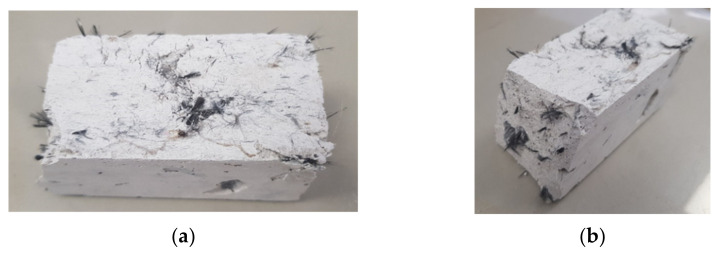
Results of the bending test, plan view (**a**); perspective view (**b**). (Source: the author.)

**Figure 3 polymers-14-01522-f003:**
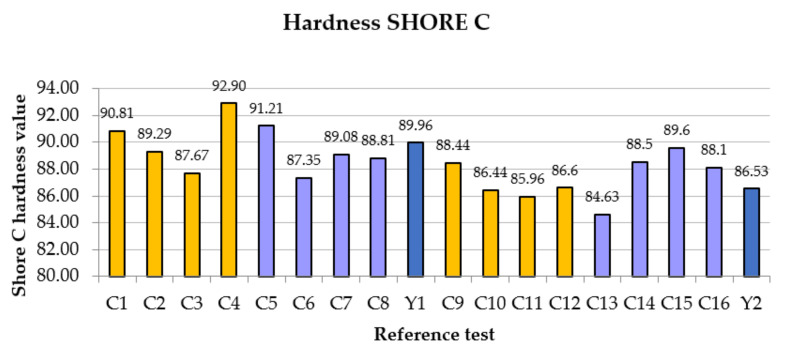
Comparative shore hardness Results.

**Figure 4 polymers-14-01522-f004:**
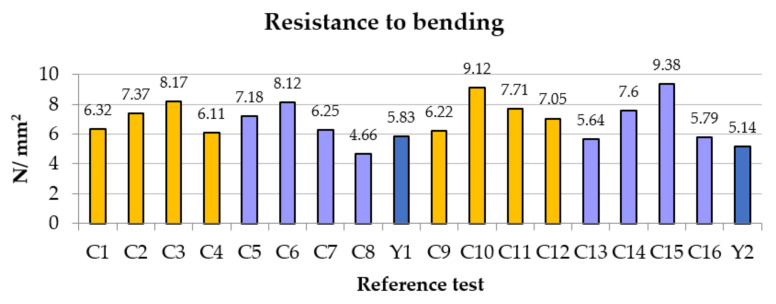
Comparative bending results.

**Figure 5 polymers-14-01522-f005:**
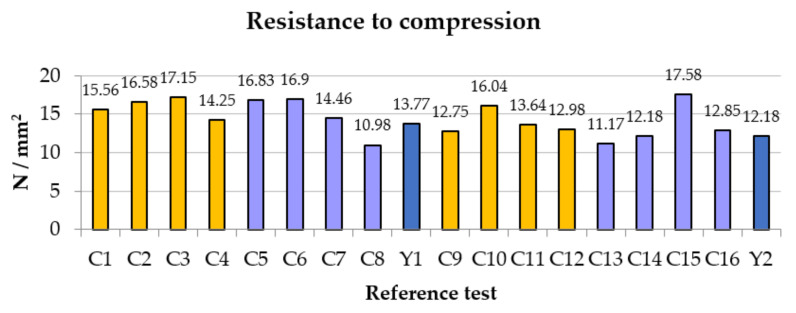
Comparative results of the compressive strength.

**Figure 6 polymers-14-01522-f006:**
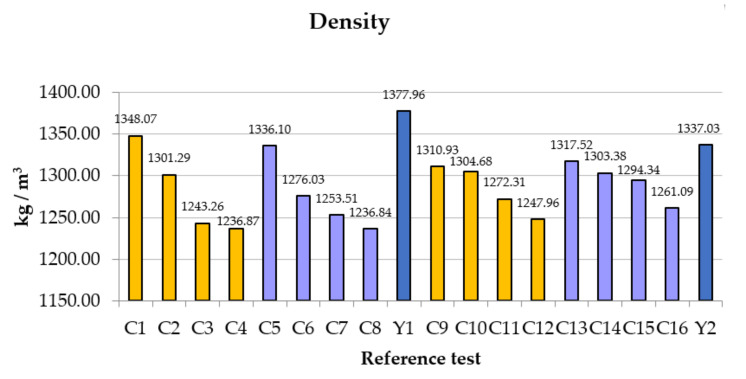
Comparative results of density.

**Figure 7 polymers-14-01522-f007:**
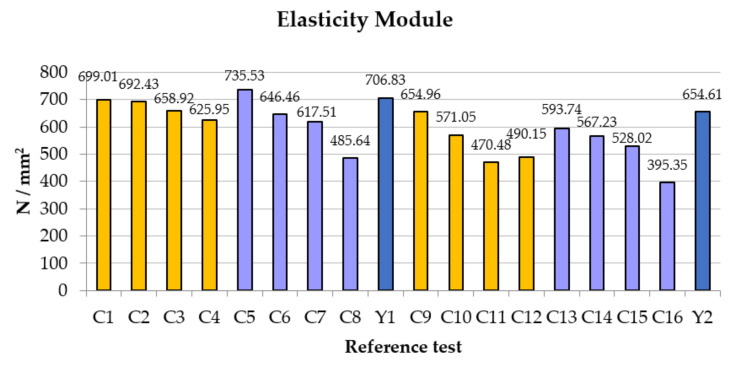
Comparative of modulus of elasticity results.

**Figure 8 polymers-14-01522-f008:**
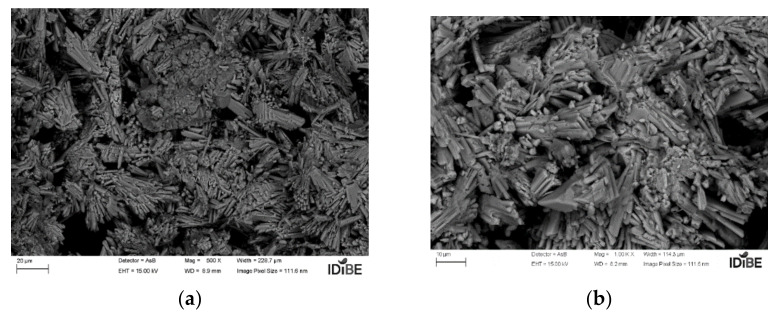
Additive plaster 1—Y1 micrograph: (**a**) 20 µm; (**b**) 10 µm (source: the author).

**Figure 9 polymers-14-01522-f009:**
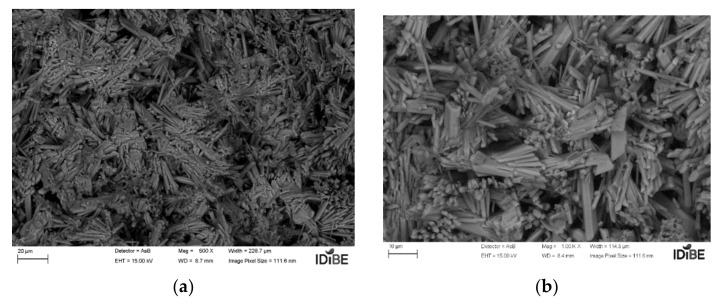
Additive plaster 1—Y2 micrograph: (**a**) 20 µm; (**b**) 10 µm (source: the author).

**Figure 10 polymers-14-01522-f010:**
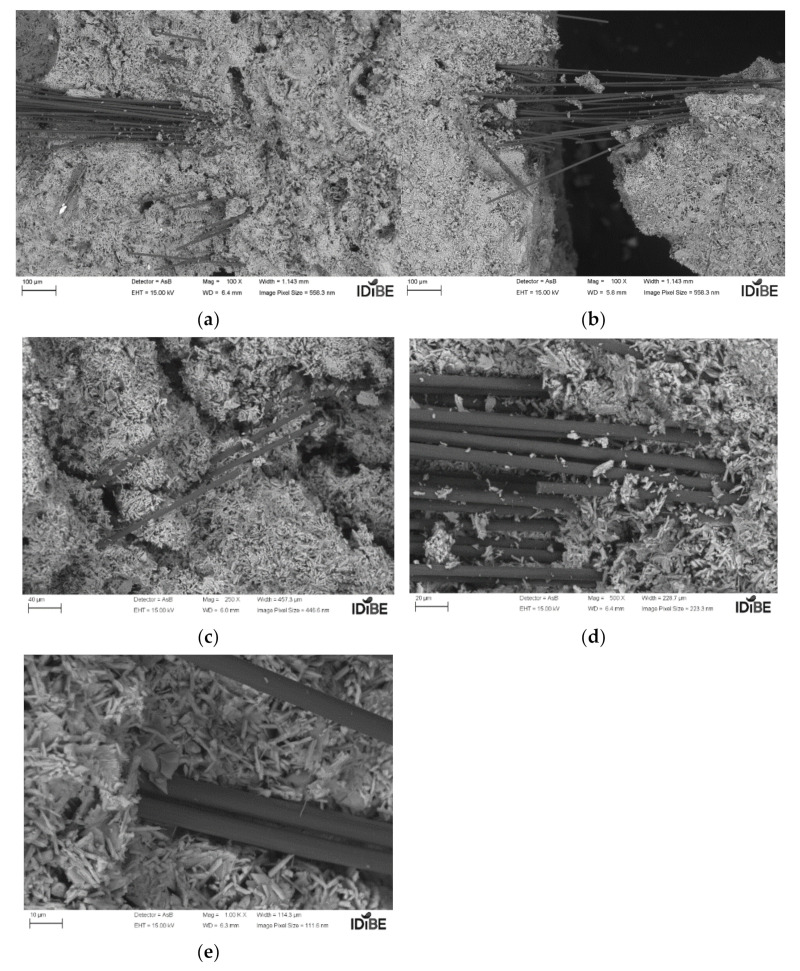
Carbon-calcium sulfate compound microscopy: (**a**) 100 µm; (**b**) 100 µm; (**c**) 40 µm; (**d**) 30 µm; (**e**) 10 µm (source: the author).

**Figure 11 polymers-14-01522-f011:**
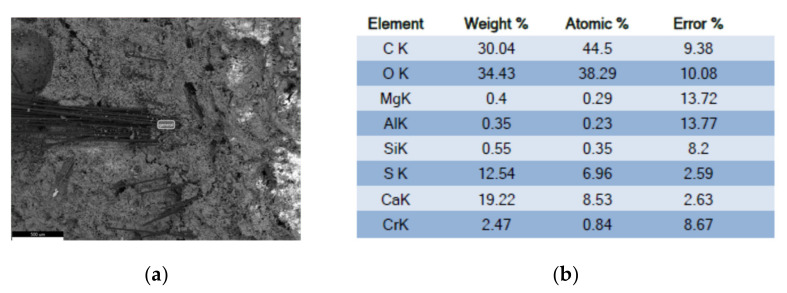
Analysis area and composition results (**a**) micrograph of calcium sulfate with additive; (**b**) chemical composition of the reference gypsum. (Source: the author.)

**Figure 12 polymers-14-01522-f012:**
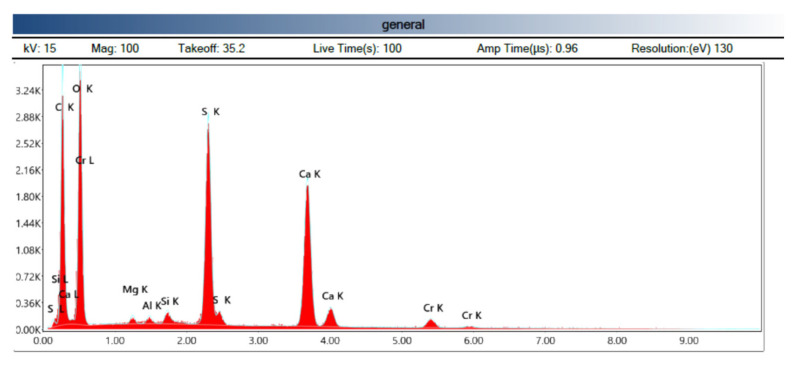
Spectrography of the chemical composition (Source: the author.)

**Figure 13 polymers-14-01522-f013:**
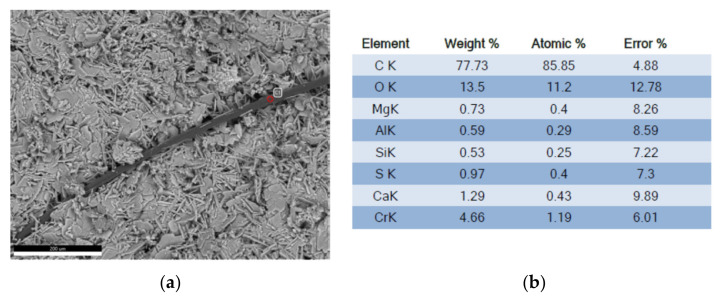
Analysis area and composition results (**a**) micrograph of calcium sulfate with additive with carbon fiber; (**b**) composition results of the area referred to carbon fiber. (Source: the author.)

**Figure 14 polymers-14-01522-f014:**
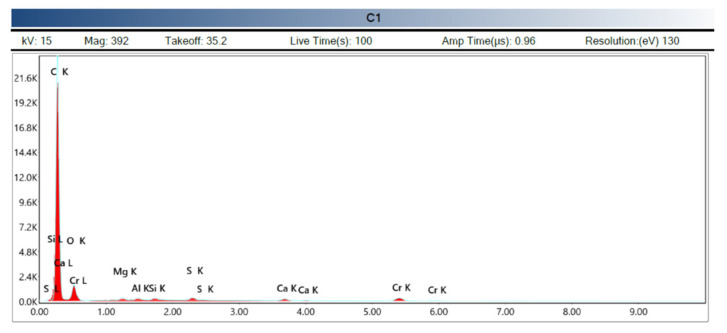
Out-of-range measurement spectrography of carbon fiber, calcium sulfate matrix. (Source: the author).

**Figure 15 polymers-14-01522-f015:**
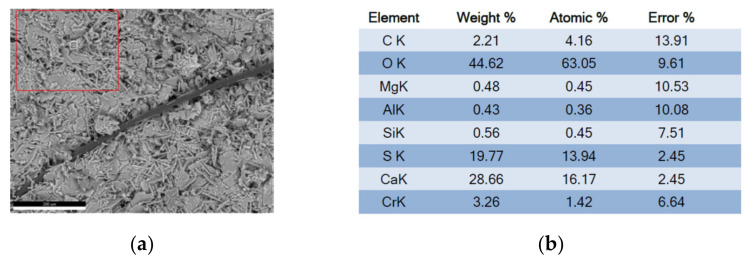
Analysis area and composition results (**a**) micrograph of the analysis area; (**b**) composition results of annexed area. (Source: the author.)

**Figure 16 polymers-14-01522-f016:**
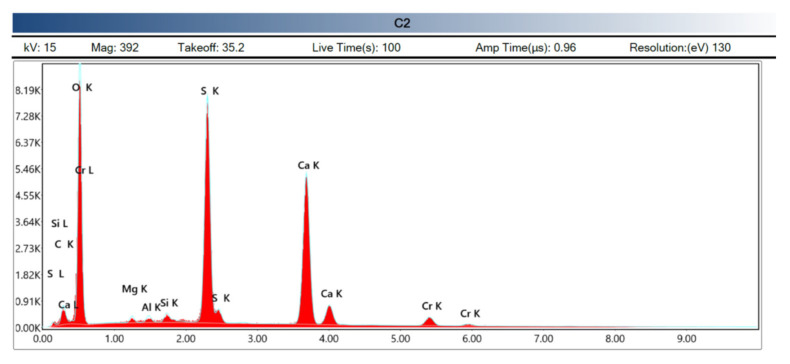
Analysis area and results for the compound (Source: the author.)

**Table 1 polymers-14-01522-t001:** Analytical parameters of the water used in the preparation of the mortars (Source: Aquagest Levante, S.A.).

Analytical Parameters of the Water Used		
Parameters	Results (%)	Units
Ammonium	<0.10 ± 15	mg/L
Total organic carbon	1.8 ± 15	mg/L
Total cyanides	<5 ± 28	μg/L
Index of Langelier	0.46	-
Bicarbonates	156.0 ± 12	mg/L
Calcium	96.4 ± 12	mg/L
Carbonates	<2.0 ± 13	mg/L
Conductivity at 20 °C	931 ± 12	μS/cm
pH	7.9 ± 0.1	pH
Sodium	54.7 ± 12	mg/L
Chlorides	90.0 ± 13.0	mg/L
Fluorides	0.150 ± 12.9	mg/L
Nitrates	2.6 ± 13.1	mg/L
Sulfates	270.8 ± 13.1	mg/L

**Table 2 polymers-14-01522-t002:** Specifications for construction plasters (source: UNE EN 13279-2) [26].

Plasters Used for Construction	Binder Plaster % Content	Principle Setting Time (min)	Flexural Strength (N/mm^2^)	Resistance to Compression (N/mm^2^)	Surface Hardness (N/mm^2^)	Adhesion (N/mm^2^)	
		Normal Application Plaster	Plaster Projection Mechanics				
1	≥50	>20 ^b^	>50	≥1.0	≥2.0	-	The break occurs on the stand or the mass of plaster. When the break appears in the plaster support, the interface must be ≥1
B2	<50						
B3	^a^						
B4	≥50						
B5	<50						
B6	^a^						
B7	≥50			≥2.0	≥6.0	≥2.5	

^a^ According to paragraphs 3.3, 3.4, 3.5, 3.6.; ^b^ some handheld applications allow one value less than 20 min. In such cases, the producer must declare the principle setting time.

**Table 3 polymers-14-01522-t003:** General properties of the additive.

Properties	Description
Additive Type	Citric acid in liquid solution
Shape	liquid
Color	Translucent—reddish
pH	3
Density (20 °C)	1.020 g/L
Viscosity at 28 °C	0.000833 kg/(m·s)
Reactivity	immediate

**Table 4 polymers-14-01522-t004:** General properties of carbon fiber.

Description	Values
Number of filaments	3 K
Weight per meter of thread:	0.2 g
Thread thickness	0.2 mm
Tensile strength	3950 N/mm^2^
Traction modulus	238 GPa
Elongation	1.7%
Density	1.76 g/cm^3^
Filament diameter	7 μ
Electrical resistivity	1.6 × 10^−3^ ohms·cm

**Table 5 polymers-14-01522-t005:** Sizes, proportions and amount of fiber per 2500 g of plaster.

Fiber Length	Fiber Added Percentage (%)	Weights (g)	Assay Reference
12 mm	0.25	6.25	C1
	0.50	12.50	C2
	0.75	18.75	C3
	1.50	37.50	C4
25 mm	0.25	6.25	C5
	0.50	12.50	C6
	0.75	18.75	C7
	1.50	37.50	C8
32 mm	0.25	6.25	C9
	0.50	12.50	C10
	0.75	18.75	C11
	1.50	37.50	C12
50 mm	0.25	6.25	C13
	0.50	12.50	C14
	0.75	18.75	C15
	1.50	37.50	C16

**Table 6 polymers-14-01522-t006:** Quantity of dosages of the different materials used in the mixture.

Ref. Contrast	Ref. Test	Additive (%)	Additive by Weight (g)	Calcium Sulfate (g)	Water (g)
Y1	C1–C8	0.02	0.5	2500	1250
Y2	C9–C16	0.06	1.5	2500	1500

**Table 7 polymers-14-01522-t007:** Results of tests of elasticity, bending, hardness and compression of the batches.

Test	Density (kg/m^3^)	Stress (N/mm^2^)	Elasticity M (N/mm^2^)	Shore C	Compression (N/mm^2^)
C1—0.25%	1348.07 ± 6.41 (l)	6.32 ± 0.30 (cde)	699.01 ± 186.24 (fgh)	90.81 ± 1.63 (hi)	15.56 ± 1.56 (cde)
C2—0.50%	1301.29 ± 6.88 (g)	7.37 ± 0.42 (fg)	692.43 ± 121.07 (fgh)	89.29 ± 1.37 (efgh)	16.58 ± 0.19 (abc)
C3—0.75%	1243.26 ± 14 (b)	8.17 ± 1.11 (g)	658.92 ± 65.32 (efgh)	87.67 ± 2.63 (bcde)	17.15 ± 1.01 (a)
C4—1.50%	1236.87 ± 7.56 (a)	6.11 ± 0.73 (cd)	625.95 ± 113.62 (defg)	92.9 ± 1.69 (j)	14.25 ± 1.68 (e)
C5—0.25%	1306.1 ± 16.34 (hi)	7.18 ± 0.76 (ef)	735.53 ± 130.57 (h)	91.21 ± 2.54 (i)	16.83 ± 2.69 (de)
C6—0.50%	1276.03 ± 45.07 (e)	8.12 ± 1.53 (g)	646.46 ± 173 (defg)	87.35 ± 3.65 (bcd)	16.90 ± 2.21 (abc)
C7—0.75%	1253.51 ± 51.96 (c)	6.25 ± 0.56 (cd)	617.51 ± 128.86 (def)	89.08 ± 6.42 (defgh)	14.46 ± 1.05 (a)
C8—1.50%	1236.84 ± 19 (a)	4.66 ± 0.4 (a)	485.64 ± 130.43 (b)	88.81 ± 1.96 (defg)	10.98 ± 0.93 (a)
Y1	1377.96 ± 14.74 (m)	5.83 ± 0.54 (bc)	706.83 ± 40.98(gh)	89.96 ± 0.69 (ghi)	13.77 ± 0.54 (abc)
C9—0.25%	1310.93 ± 8.48 (i)	6.22 ± 0.88 (cd)	654.96 ± 97.62 (efgh)	88.44 ± 3.52 (defg)	12.75 ± 0.38 (abc)
C10—0.50%	1304.68 ± 19.83 (gh)	9.12 ± 1.33 (h)	571.05 ± 79.23 (cd)	86.44 ± 4.46 (bc)	16.04 ± 1.19 (bcd)
C11—0.75%	1272.31 ± 11.43 (e)	7.71 ± 0.68 (fg)	470.48 ± 77.3 (b)	85.96 ± 4.33 (ab)	13.64 ± 3.65 (abc)
C12—1.50%	1247.96 ± 7.87 (b)	7.05 ± 0.45 (def)	490.15 ± 200.51 (b)	86.6 ± 3.17 (bc)	12.98 ± 1.60 (bcd)
C13—0.25%	1317.52 ± 18.12 (j)	5.64 ± 0.5 (bc)	593.74 ± 86.01 (cde)	84.63 ± 3.58 (a)	11.17 ± 1.12 (a)
C14—0.50%	1303.38 ± 21.52 (gh)	7.6 ± 0.84 (fg)	567.23 ± 119.87 (cd)	88.5 ± 3.82 (defg)	12.18 ± 0.91 (ab)
C15—0.75%	1294.34 ± 45.68 (f)	9.38 ± 1.96 (h)	528.02 ± 95.03 (bc)	89.6 ± 2.56 (fghi)	17.58 ± 1.36 (de)
C16—1.50%	1261.09 ± 14.11 (d)	5.79 ± 0.66 (bc)	395.35 ± 83.9 8 (a)	88.1 ± 2.64 (cdef)	12.85 ± 0.57 (abc)
Y2	1337.03 ± 7.81 (k)	5.14 ± 0.36(ab)	654.61 ± 39.31 (efgh)	86.53 ± 0.48 (bc)	12.18 ± 0.66 (ab)

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
