# Peer review of "Mechanical Behavior of Calcium Sulphate Modified with Citric Acid and with Added Carbon Fibers"

_polymers, 2022, doi:10.3390/polym14081522_

Round 1

Reviewer 1 Report

This a well-written manuscript which I am happy to recommend for publication in Polymers.

Author Response

Dear Reviewer
Thank you very much for your excellent comments, I am very happy that you have liked this article.
I hope to have you for future article reviews
Thank you so much
Best regards
Dr. Luis Serna

Reviewer 2 Report

paper dealing with MECHANICAL BEHAVIOR OF CALCIUM SULPHATE MODIFIED WITH CITRIC ACID ANDADDED WITH CARBON FIBERS. Although the topic is interesting, the paper needs major revision and several point for improvement before publication.

1- abstract: please follow the journal style in writing this section. The abstract must shows the novelty of the study and the strong conclusion and results. 

2- The introduction: this section must be improved. must add new 10-15 ref. to show the importance of this study and what can be improved thought the properties of additive and CF (carbon fiber).

what is the novelty of this study?? must show it.

3- The methods of materials and testing are ok, however, adding a "flow chart" is must to improve this section.

4- the discussion, I'm really confused as the authors keep it without given the figure no. [ all figures???] , for instance line 232-235 " As the water increases, 232
the hardness decreases significantly. It is already observed with the test reference Y1 233
(89.96) and Y2 (86.53) that they do not contain carbon fibers. On the other hand, the deviations in the tests depend on the fact that, when carrying out the measurement, there are certain "holes" caused in the kneading by the fibers. " which Figure?????!!! . and this applied to all figures and discussion section. This is must fix.

5- Conclusion: needs improvement to show the research gap and the findings of this current study. This must improved. 

what is the future recommendation????  must add.

Finally, there are several grammar issues, and many misuse in the english letters, for instance,

line 33: System? make it system. (small letters).

line 35 : Climate Change? make it climate change

line 105: an addition? make it an additive.

line 107-108: Shore surface hardness, Density, Flexural strength. , Compression, Elasticity and microscopic matrix. ? make it shore surface hardness, density, flexural strength. , compression, elasticity and microscopic matrix .

line 223: Below are the comparative tables in relation to the length? which tables ? give no.

each figure must has no. in the discussion and the discussion must compare the current results with other studies. this must add in the new discussion.

Author Response

Dear Reviewer
Thank you very much for your comments, I have made the corrections you have indicated, I hope you like them.
Below, he answered about his doubts

1- abstract: please follow the journal style in writing this section. The abstract must shows the novelty of the study and the strong conclusion and results. 

It has been corrected, as a summary and adjusting to the 200 words established in the editorial regulations, the novelty of the study has been included, the conclusions and the results are solid.

2- The introduction: this section must be improved. must add new 10-15 ref. to show the importance of this study and what can be improved thought the properties of additive and CF (carbon fiber).

what is the novelty of this study?? must show it.

The novelty of this study is the inclusion of polymeric fibers in a material used over the centuries to improve its mechanical properties, with these improvements we will be able to reduce thicknesses in manufacturing, which implies a reduction in manufacturing energy and weight. structures in buildings, which should be studied and analyzed in the future.

3- The methods of materials and testing are ok, however, adding a "flow chart" is must to improve this section.

As you well indicate, the experimentation method that has been carried out is described. The flowchart, as you well know, is the graphic representation of a process or algorithm, that is, the graphic summary of what we have done; I take note for future articles, and I correct as you indicate and I add the flowchart.

4- the discussion, I'm really confused as the authors keep it without given the figure no. [ all figures???] , for instance line 232-235 " As the water increases, 232
the hardness decreases significantly. It is already observed with the test reference Y1 233
(89.96) and Y2 (86.53) that they do not contain carbon fibers. On the other hand, the deviations in the tests depend on the fact that, when carrying out the measurement, there are certain "holes" caused in the kneading by the fibers. " which Figure?????!!! . and this applied to all figures and discussion section. This is must fix.

Corrected

5- Conclusion: needs improvement to show the research gap and the findings of this current study. This must improved. 

what is the future recommendation????  must add.

The future recommendation is the study of the energy quantification of the manufacturing process, the energy savings that are produced, and the reduction of carbon dioxide emissions into the atmosphere.

Finally, there are several grammar issues, and many misuse in the english letters, for instance,

line 33: System? make it system. (small letters).Corrected

line 35 : Climate Change? make it climate change. Corrected

line 105: an addition? make it an additive. Corrected

line 107-108: Shore surface hardness, Density, Flexural strength. , Compression, Elasticity and microscopic matrix. ? make it shore surface hardness, density, flexural strength. , compression, elasticity and microscopic matrix .Corrected

line 223: Below are the comparative tables in relation to the length? which tables ? give no. Corrected

Thank you very much
All the best
Dr. Luis Serna.

Round 2

Reviewer 2 Report

accepted for publication in the current version.

This manuscript is a resubmission of an earlier submission. The following is a list of the peer review reports and author responses from that submission.

Round 1

Reviewer 1 Report

The work entitled “MECHANICAL BEHAVIOR OF CALCIUM SULPHATE MODIFIED WITH CITRIC ACID AND ADDED WITH CARBON FIBERS” reports on the impact of using a combination of citric acid and carbon fibers of different sizes in the mechanical resilience of calcium sulfate. Even though the manuscript is deals with a subject that is pertinent, its quality is very low. It is imperative that the information provided is well funded and targets an actual need. There are many details that require the authors’ attention prior to being considered for publication:

  • This is a very poorly written manuscript. The English writing is extremely deficient.
  • The introduction is very superficial. From what is provided we cannot see the actual impact of this work, how it improves on other publication of its kind and what are the major novelties introduced.
  • Table should provide the same number of decimal cases per information. In English, the decimal case is indicated with a “.” and not “,”.
  • Why would we need Figure 3?
  • Histograms are in some instances outside the margins and do not possess standard deviation.
  • There is very little discussion of the data. The authors did mostly the presentation of the results than a proper discussion, based on a critic overview of the data that is improved by the presence of literature work.

Reviewer 2 Report

  1. Abstract requires rewriting. It does not follow a logical sequence.
  2. There is a lot of detail in the description of the experimental methodology which is good.  However, some of the terms used such as test-tubes (instead of samples) need correction.
  3. A tow with 3K filaments is normal in carbon fibers.  It appears that during manual kneading, the fibers were not separated and appear to form clumps. Was any orientation of the fibers envisioned or were the fibers randomly oriented? What was the choice of length of fibers and their volume fraction based upon? The volume fractions (0.25%, 0.50%, 0.75% and 1.5%) appear to be too low to make a significant impact on the properties of the composite.
  4. Was any sizing used to improved fiber-matrix adhesion?
  5. The images in Fig.1, Fig. 2a, Fig. 3a and 3b can be eliminated.
  6. The tables and figures do not have proper captions.  Some more details are necessary in the captions.
  7. The values of the mechanical properties do not show a consistent trend. What do the terms "Stress", "Shore C"  and "Compression" refer to? Do they mean Flexural of Tensile Strength, Hardness and Compressive Strength? Does Elasticity M refer to Modulus of Elasticity?
  8. In line 248, it is difficult to understand "the greatest deformation" is obtained with the "small values of carbon fiber".  Please clarify.
  9. It appears that Figs. 5a and 5b are different magnifications of the same composite.  What are the different features observed from the two different magnifications? The same is true for Figs. 6a and 6b. Fig. 7 also has many images without clarification for their use in the manuscript.
  10. What is the purpose of using EDS?
  11. There are two Fig. 11 and Figures 10-12 are not cited in the text.
  12. It appears that the improvement in the properties is 82% instead of 182% and 44% instead of 144%. 
  13. The last two paragraphs in the conclusion section are not from the experimental work described.
  14. References appear to be from a very limited circle.